# MENTALARENA: SELF-PLAY TRAINING OF LANGUAGE MODELS FOR DIAGNOSIS AND TREATMENT OF MENTAL HEALTH DISORDERS

## ABSTRACT

Mental health disorders are one of the most serious diseases in the world. Most people with such a disease lack access to adequate care, which highlights the importance of training models for the diagnosis and treatment of mental health disorders. However, in the mental health domain, privacy concerns limit the accessibility of personalized treatment data, making it challenging to build powerful models. In this paper, we introduce *MentalArena*, a self-play framework to train language models by generating domain-specific personalized data, where we obtain a better model capable of making a personalized diagnosis and treatment (as a therapist) and providing information (as a patient). To accurately model human-like mental health patients, we devise *Symptom Encoder* which simulates a real patient from both cognition and behavior perspectives. To address intent bias during patient-therapist interactions, we propose *Symptom Decoder* to compare diagnosed symptoms with encoded symptoms, and dynamically manage the dialogue between patient and therapist according to the identified deviations. We evaluated MentalArena against 6 benchmarks, including biomedicalQA and mental health tasks, compared to 6 advanced models. Our models, fine-tuned on both GPT-3.5 and Llama-3-8b, significantly outperform their counterparts, including GPT-4o. We hope that our work can inspire future research on personalized care.

## 1 INTRODUCTION

Mental health disorders include a variety of conditions such as anxiety, depression, and schizophrenia, which affect people's thinking, emotions, behavior, or mood (Prince et al., 2007). In 2019, approximately 970 million people worldwide lived with a mental health disorder, with anxiety and depression being most prevalent (WHO, 2022). The number increased by 28% in 2020 and continues to increase. Despite the availability of effective treatments, many individuals lack access to adequate care due to under-resourced health systems. For example, only 29% of people with psychosis and one third of people with depression receive formal mental healthcare (WHO, 2022). It is indispensable to develop machine learning models for the automatic diagnosis and treatment of such diseases. However, existing AI therapist systems use templates and decision trees, which are not flexible to support personalized care (Fiske et al., 2019; D'Alfonso, 2020; Grodniewicz & Hohol, 2023).

The key to training powerful models is to collect sufficient training data. However, due to privacy concerns in the medical domain, data collection, especially personalized data for mental health disorders, is inherently challenging. A growing body of work has focused on enhancing mental health language models by sourcing additional domain-specific data from social media (Xu et al., 2024; Yang et al., 2024a; Hu et al., 2024a). However, social media data are inherently biased and under-representative, failing to capture the full spectrum of people's mental health needs. Moreover, as LLMs continue to scale, the availability of training data in the real world becomes increasingly limited, further exacerbating this challenge. Existing methods are likely to soon reach their performance limit.

Recently, several works have focused on self-play (Hu et al., 2024b; Yang et al., 2024b; Liang et al., 2024; Wu et al., 2024; Wang et al., 2024d), where models play different roles and self-evolve or co-evolve during interaction with other models. A model synthesizes training data on its own and then use the generated data to train itself. However, there are two challenges that prevent us from

Figure 1: MentalArena is a self-play framework for the diagnosis and treatment of mental health disorder consisting of three modules: *Symptom Encoder*, *Symptom Decoder*, and *Model Optimizer*.

adopting self-play training for mental health disorders: (1) Scarcity of high-quality data. Since mental health disorder is a complicated disease that involves symptoms of cognition and behavior, current LLMs lacks such a personalized experience to accurately simulate patients with different conditions (Schmidgall et al., 2024; Wang et al., 2024a). (2) Intent bias. Intent bias often occurs, where the "patient" expresses one view, but the "therapist" misinterprets it due to knowledge gaps, mirroring real therapist-patient misunderstandings (Britten et al., 2000; West, 1984; Shreevastava & Foltz, 2021).

MentalArena is a framework specifically designed for self-play training of language models to facilitate the diagnosis, treatment, and medication of mental health disorders. The model $M$ assumes the dual roles of both patient and therapist. In its capacity as the therapist, it provides diagnoses, treatment plans, and medication regimens based on the symptoms presented by the patient. As the patient, it simulates its updated health status after implementation of each treatment and medication plan. As illustrated in Figure 1, MentalArena comprises three key modules: *Symptom Encoder*, *Symptom Decoder*, and *Model Optimizer*. *Symptom Encoder* models mental health patients based on cognitive models [1] and behavioral patterns, offering rich insights into coping strategies and behavioral principles. *Symptom Decoder* simulates the diagnosis and treatment interactions between a patient and a therapist, generating more personalized dialogues while mitigating intent bias (Britten et al., 2000; West, 1984). During each iteration, we collect data from these interactions, including diagnostic, treatment, and medication information, and evolve the models through training on those datasets.

To evaluate MentalArena, we conduct experiments on 6 benchmarks including datasets on biomedical QA and mental health detection. We compare our fine-tuned models with other state-of-the-art and mental health models. We also compare with two advanced prompt engineering approaches. Our models outperform all their counterparts. Specifically, MentalArena brings a great improvement to base models (20.7% improvement over GPT-3.5-turbo and 6.6% over Llama-3-8b). Moreover, our model based on GPT-3.5-turbo significantly outperforms GPT-4o by 7.7%.

We further thoroughly analyze the dynamics of self-play training. We find that the perplexity score (Marion et al., 2023; Wang et al., 2023) and the model performance are highly correlated. For diversity gain (Bilmes, 2022), the model performance will increase if the diversity gain exceeds some thresholds. We also explore whether MentalArena can be generalized to other diseases. The results on MedMCQA (Pal et al., 2022) and MMLU (Hendrycks et al., 2020) prove the generalization ability of MentalArena in medical domain. Furthermore, we explore the catastrophic forgetting of our fine-tuned models. The results on BIG-Bench-Hard (BBH) (Suzgun et al., 2022) show that our models does not decrease performance in general benchmarks and can even improve their results.

In summary, the contributions of this paper are following:

1. We propose MentalArena, a novel and cost-effective self-play framework for training language models for diagnosing and treating mental health disorders. MentalArena introduces *Symptom Encoder* and *Symptom Decoder*, designed to simulate real patient-therapist interactions by modeling cognitive and behavioral processes.

2. Using MentalArena, we generate high-quality data containing diagnosis, treatment, and medication data. There are $18k$ samples in total that can be used for further training and research.

---

[1]The cognitive model is designed based on cognitive behavior therapy (CBT) principles (Beck, 2020), a popular paradigm in psychotherapy. Appendix A.6.1 shows the example of cognitive models.

3. We evaluate MentalArena on 6 benchmarks comparing with 6 LLMs. Our models based on GPT-3.5-turbo and Llama that are trained through the MentalArena framework outperform all off-the-shelf counterparts, including GPT-4o.

## 2 RELATED WORK

### 2.1 LARGE LANGUAGE MODELS FOR HEALTHCARE

Researchers have explored the potential of large language models (LLMs) in healthcare (Jiang et al., 2023; Li et al., 2023; Liu et al., 2023; Lupetti et al., 2023; Nori et al., 2023a; Singhal et al., 2023; Wu et al., 2023; Wang et al., 2024c). For example, Singhal et al. (2023) fine-tuned PaLM-2 for medical applications, achieving 86.5% accuracy on the MedQA dataset. Similarly, Wu et al. (2023) fine-tuned LLaMA on medical literature, showing strong performance in biomedical QA tasks.

In the mental health domain, research has taken two main approaches. The first involves fine-tuning domain-specific LLMs on existing datasets or social media data, such as Mental-LLaMA (Yang et al., 2024a) and Mental-LLM, fine-tuned on Reddit data (Xu et al., 2024). The second approach enhances mental health performance through prompt engineering. Yang et al. (2023) proposed emotion-enhanced prompting strategies to guide LLMs in explainable mental health analyses.

Unlike previous methods, MentalArena fine-tunes mental health models through self-play training, in which the base model assumes both patient and therapist. Training data is generated dynamically during the interactions between these two roles, allowing for more effective model refinement.

### 2.2 SELF-PLAY FRAMEWORKS IN LARGE LANGUAGE MODELS

Self-play involves a model evolving through interactions with copies of itself, creating a feedback loop that refines performance without external input. It is particularly effective in environments where the model simulates multiple roles, such as multiplayer games (Silver et al., 2016; 2017). Compared to interactive methods, self-play provides a more efficient strategy for obtaining feedback without relying on an external environment.

Taubenfeld et al. (2024) examine biases in LLM-generated debate simulations, while Ulmer et al. (2024) focus on principle-guided conversations. Role-playing approaches, like Lu et al. (2024)'s self-simulated dialogues with character profiles and Askari et al. (2024)'s SOLID framework for intent-aware role-play, leverage LLMs to generate information-rich exchanges.

Due to the lack of sufficient data in the training corpus, LLMs are unable to accurately simulate real patients, presenting a significant challenge for self-play training. To overcome this, MentalArena introduces *Symptom Encoder*, a component designed to effectively model real mental health patients.

## 3 MENTALARENA

### 3.1 PRELIMINARIES

We first go over the process of the diagnosis and treatment of mental health disorder and explain key concepts. Mental health diagnosis begins with assessing an individual's *health state*, encompassing mental and emotional well-being. *Symptoms* are key indicators of possible problems, including emotional (e.g., anxiety, depression), cognitive (e.g., memory problems) and behavioral changes (e.g., social withdrawal). These symptoms lead to a formal *diagnosis* made through clinical interviews identifying specific disorders such as depression, anxiety, or schizophrenia. Once diagnosed, the *treatment* process begins, often involving a combination of psychotherapy (e.g. cognitive-behavioral therapy), lifestyle changes, and sometimes medication. *Medications*, such as antidepressants and mood stabilizers, are used to regulate brain chemicals and alleviate symptoms. (Prince et al., 2007)

### 3.2 OVERVIEW OF THE FRAMEWORK

Although it is trivial to adopt the self-play training paradigm in fine-tuning general language models (Taubenfeld et al., 2024; Ulmer et al., 2024; Lu et al., 2024; Askari et al., 2024; Wang et al.,

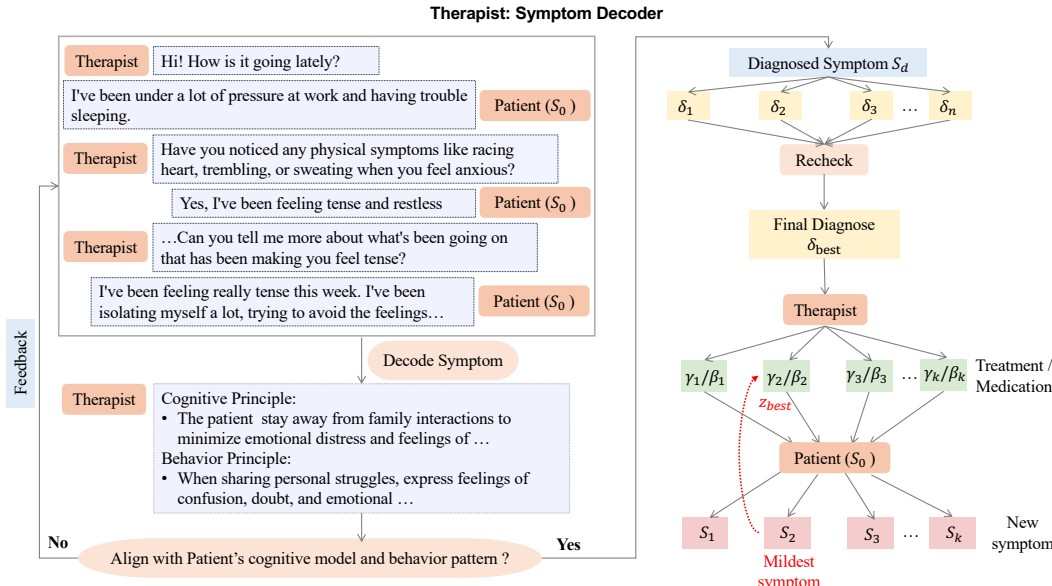

Figure 2: Symptom Decoder aims to mitigate the intent bias between therapists and patients through patient decoding and dynamic control of the conversation. To ensure the accuracy of the diagnostic information provided by the therapist, the patient simulates their updated health condition after implementing the prescribed treatment or medication plan.

2024b;d), it remains unexplored and challenging to exploit such a framework in the medical domain due to the data deficiency in medical and intent bias problem between patients and therapists. The first challenge makes it difficult to play a patient role (Schmidgall et al., 2024; Wang et al., 2024a) due to the data deficiency of the patient in the training corpus, while the latter undermines the effective diagnosis and treatment of explicit symptoms.

MentalArena is a framework designed specifically for self-play training of language models to facilitate the diagnosis, treatment and medication of mental health disorder. As shown in Figure 1, MentalArena consists of three key modules: *Symptom Encoder*, *Symptom Decoder*, and *Model Optimizer*. Specifically, *Symptom Encoder* is designed to model mental health patient from cognitive models and behavioral patterns, providing a wealth of information on the coping strategy and behavior principles. *Symptom Decoder* emulates the process of diagnosis and treatment between a patient and a therapist to generate a more personalized dialogue while mitigating intent bias (Britten et al., 2000; West, 1984). At each iteration, we collect the data during interactions, including diagnosis data, treatment data, and medication data, and evolve the models via training on those datasets.

Formally, we use $\mathbf{x}$ to denote the initial health information of a patient and $M$ to denote the base model (e.g., GPT-3.5) for the therapist and the patient via role-play strategy. Our objective is to obtain $M^*$ via self-play training that can achieve better performance in both personalized diagnosis and treatment of the patient (as a therapist) and information disclosure (as a patient). Self-play training is conducted taking as input both original information $\mathbf{x}$ and treatment or medication information $\mathbf{z}$ generalized by $M$. In iteration $t$, the model $M_t$ plays the therapist $D_t = M_t(\cdot \mid \text{Prompt}_{doc})$ and the patient $P_t = M_t(\cdot \mid \text{Prompt}_{pat})$, which generates diagnosis and treatment data (Figure 8) during patient-therapist interactions. The module *Symptom Encoder* can be seen as learning the encoded symptom $S_0$ by disentangling the initial health information $\mathbf{x}$ into cognitive and behavioral principles.

Then, the module *Symptom Decoder* generates a personalized dialogue containing key information $\mathbf{z} = \{\delta, \beta, \gamma\}$, where $\delta, \beta$ and $\gamma$ denote the diagnosis, treatment and medication of the patient given the symptom $S_0$. It consists of $k$ rounds of communication in which the patient can provide more accurate and sufficient information by accepting the treatment and medication given by the therapist in each round. As treatment and medication plans are administered to the patient, their health state evolves, reflected in the sequential updates of encoded symptoms, denoted as $S_1, S_2, ..., S_{k-1}$. The encoded symptoms serve as indicators of the effectiveness of the treatment and medication plans,

progressively updating as interventions are carried out. Eventually, the therapist will provide the optimal diagnosis information $\mathbf{z}_{best} = \{\delta_{best}, \beta_{best}, \gamma_{best}\}$ which is crucial for model optimization[2].

Finally, in *Model Optimizer*, we fine-tune the model using the paired data $(S_1, \delta_{best}), (S_d, \gamma_{best})$, and $(S_d, \beta_{best})$. This iterative training requires $T$ rounds to obtain the optimal model.

### 3.3 PATIENT: SYMPTOM ENCODER

The module *Symptom Encoder* aims to model mental health patient from both cognitive and behavioral perspectives, which learns meaningful symptoms $S_0$ from the original patient health data $\mathbf{x}$. Specifically, the module learns symptoms from the aspects of cognition and behavior. The cognitive model is designed based on cognitive behavior therapy (CBT) principles (Beck, 2020), a popular paradigm in psychotherapy. Cognitive models address maladaptive cognitive structures that are embedded in various contexts, including familial conflicts, relationship challenges, workplace challenges, and other areas. The models consist of eight key components: relevant history, core beliefs, intermediate beliefs, coping strategies, situational factors, automatic thoughts, emotions, and behaviors. (Beck, 2020) The explanation of each component of the cognitive model can be found in Appendix A.6.2. Appendix A.6.1 shows the example of cognitive models. We obtain 106 patient cognitive models from previous work (Wang et al., 2024a), which are created by clinical psychologists. To simulate the cognitive activity of the mental health patient, we encode those cognitive models into patient via prompt. Our prompts are shown in Appendix A.1.

For patient behavior modeling, we use behavior principles collected by Louie et al. (2024) as a behavior library, created by 25 mental health experts. Examples of behavior patterns are shown in Appendix A.6.1. To find the proper behavior pattern for each cognitive model, we first semantically match the coping strategies of cognitive model with each behavior pattern. We obtain the embeddings for each coping strategy and behavior principle via Bert-base (Devlin et al., 2018), considering on effectiveness and cost. Then we compute the semantic similarity between coping strategies and behavior pattern. The max similarity score of all behavior principles in one behavior pattern is selected to represent the score of the pattern. The five behavior patterns with the highest scores are kept. To further ensure the most appropriate pattern, we prompt GPT-4-turbo (OpenAI, 2023b) to pick one from the five patterns. The final behavior pattern is also integrated into patient via prompt, which is shown in Appendix A.1.

### 3.4 THERAPIST: SYMPTOM DECODER

During interactions between a real therapist and a real patient, the patient may try to express one opinion while the therapist misunderstands the intent due to prior knowledge and deficiency of experience (Britten et al., 2000; West, 1984). Intention bias can similarly arise in conversations between patients and therapists played by AI models, resulting in inaccurate diagnosis and treatment. *Symptom Decoder* is designed to mitigate the intent bias. After several conversations, the therapist reviews the patient's health information from previous interactions and conducts a detailed analysis of the patient's cognitive and behavioral patterns, resulting in the diagnosed symptom $S_d$. We then semantically match the encoded symptom $S_0$ with the diagnosed symptom $S_d$ and guide subsequent conversations based on the differences between $S_0$ and $S_d$.

As shown in Figure 2(left), the therapist decodes cognitive and behavior principles according to the conversation history. For example, the decoded cognitive principle is: "The patient stay away from family interactions to minimize emotional distress and feelings of abandonment". The decoded behavior principle is: "When sharing personal struggles, express feelings of confusion, doubt, and emotional turmoil to convey a sense of vulnerability and authenticity". Then we compute the semantic similarity score of the decoded symptom $S_d$ and the encoded symptom $S_0$. If the score is greater than $0.9$, the conversation will end, indicating that the therapist has fully understood the health state of the patient. Otherwise, it indicates the existence of intent bias. To help the therapist better know more about the health state of the patient, we summarize the differences between the decoded symptom $S_d$ and encoded symptom $S_0$ and generate some feedback for further inquiries via the

---

[2]$\delta_{best}$ represents the diagnosis plan selected by the patient from several proposed options, based on their reassessment of their health status. Similarly, $\beta_{best}$ and $\gamma_{best}$ are determined based on the patient's updated encoded symptoms after the prescribed treatments and medications have been administered.

GPT-4-turbo (OpenAI, 2023b), which can remind the therapist of missing or confusing information about the patient. For instance, the feedback is like "The therapist can focus on what is going on that has been making the patient feel tense." And the conversation will not end until the similarity score between $S_d$ and $S_0$ is greater than 0.9.

Table 1: Statistics of the evaluation datasets.

| Task | Dataset | Type | #Sample |
|---|---|---|---|
| Biomedical QA | MedQA | Multi-class Classification | 173 |
| | MedMCQA | Multi-class Classification | 314 |
| | PubMedQA | Multi-class Classification | 328 |
| Depression/suicide cause detect | CAMS | Generation | 625 |
| Stress detect | Dreaddit | Binary Classification | 414 |
| Interpersonal risk factors detect | Irf | Binary Classification | 2,113 |

After the conversation ends, the therapist analyzes the patient's symptom, $S_d$, and formulates several diagnostic plans ($\delta_1, \delta_2, ..., \delta_n$). To ensure diagnostic accuracy, the patient reviews each plan and selects the most appropriate one based on their health condition. Subsequently, the therapist proposes a series of treatment and medication plans ($\{\gamma_1, \beta_1\}, ..., \{\gamma_k, \beta_k\}$) in accordance with the selected diagnosis ($\delta_{best}$). To identify the optimal treatment and medication plans, we apply each plan to the patient (initially represented by the encoded symptom $S_0$) and monitor the progression of the patient's encoded symptoms. These symptoms are updated as different plans are implemented, reflecting the patient's evolving health state. The encoded symptom is updated to $S_1, S_2, ..., S_k$ as the treatment and medication plans ($\{\gamma_1, \beta_1\}, ..., \{\gamma_k, \beta_k\}$) are administered. As illustrated in the center of Figure 1, the patient initially transmits $S_0$ to the therapist. Following the administration of treatment or medication $z_1$, the patient's encoded symptom is updated to $S_1$. Similarly, after the application of treatment or medication $z_2$, the encoded symptom is further updated to $S_2$. The encoded symptoms serve as indicators of the effectiveness of the treatment and medication plans, progressively updating as interventions are carried out. Eventually, the therapist will provide the optimal diagnosis and treatment information $\mathbf{z}_{best} = \{\delta_{best}, \beta_{best}, \gamma_{best}\}$ which is crucial for model optimization.

### 3.5 MODEL OPTIMIZER

After obtaining treatment, diagnosis, and medication through *Symptom Decoder*, we train $M$ in a self-play manner to get a better model capable of making a personalized diagnosis and treatment (as a therapist) and presenting information (as a patient). An example of such a supervised fine-tuning process is illustrated in Figure 8.

During each iteration, the patient and the therapist are powered by the same model $M$ and both get improved when $M$ is updated. While our framework is flexible to allow for different base models for the two roles, we adopt the same one due to the following reasons. First, it is intuitive that training one base model is more efficient compared to training different models. Second, and more importantly, training one base model can help reduce the knowledge gap between two roles. Two different base models can certainly exhibit knowledge gaps, and iterative training will enlarge them due to different architectures and pre-training data of the models. Appendix A.8 shows the detailed training settings.

## 4 EXPERIMENT

### 4.1 SETUP

**Datasets:** As summarized in Table 1, we adopt 6 datasets: MedQA (Jin et al., 2021), MedMCQA (Pal et al., 2022), PubMedQA (Jin et al., 2019), CASM (Garg et al., 2022), Dreaddit (Turcan & McKeown, 2019) and Irf (Garg et al., 2023). Our evaluation spans biomedical QA and mental health detection, covering knowledge on diagnosis, treatment, and medication. These datasets include general mental health tasks, such as depression/suicide, stress, and interpersonal risk factors detection, as well as real-world mental health cases. Details on the benchmarks are provided in Appendix A.3

Table 2: Main results on Accuracy (%) for MentalArena with different base models. The final five rows are either strong methods (i.e., GPT-4o) or those designed specifically for mental health.

| Model | MedQA | MedMCQA | PubMedQA | CAMS | dreaddit | Irf | AVG |
|---|---|---|---|---|---|---|---|
| MentaLLaMa-13b | 28.32 | 12.42 | 28.96 | **37.28** | 62.08 | 46.81 | 35.98 |
| Mental-LLM-alpaca | 28.32 | 12.42 | 0.00 | 29.76 | **64.98** | 51.96 | 31.24 |
| Mental-LLM-t5 | 0.00 | 0.32 | 49.09 | 27.04 | 63.29 | 47.70 | 31.24 |
| GPT-4o | 87.86 | 74.20 | 60.06 | 27.68 | 49.03 | **64.65** | 60.58 |
| GPT-4o+MedPrompt | **90.17** | **78.34** | **67.38** | 31.52 | 53.27 | **64.65** | **64.22** |
| Base: GPT-3.5-turbo | 64.16 | 33.76 | 44.68 | 28.96 | **49.03** | **64.65** | 47.54 |
| +Chain-of-thought | 65.90 | 37.97 | 45.73 | 29.92 | **49.03** | **64.65** | 48.87 |
| +MedPrompt | 69.94 | 43.89 | 47.26 | 30.2 | **49.03** | **64.65** | 50.83 |
| +Ours | **74.57** | **91.08** | **97.56** | **32.80** | **49.03** | **64.65** | **68.28** |
| Base: Llama-3-8b | 70.52 | 42.04 | 86.59 | 25.12 | 58.45 | 45.76 | 54.75 |
| +Chain-of-thought | 75.14 | 47.77 | 88.21 | 33.6 | 62.22 | 45.91 | 58.81 |
| +MedPrompt | 76.88 | 49.41 | 89.99 | **35.08** | 61.59 | 48.05 | 60.17 |
| +Ours | **78.03** | **50.32** | **92.68** | 29.60 | **65.46** | **52.25** | **61.39** |

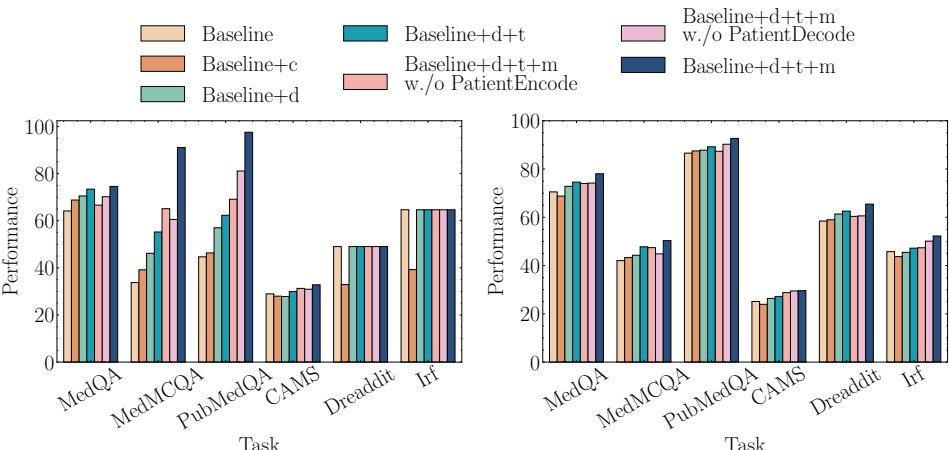

Figure 3: Ablation study. Each bar represents the performance of model trained on different settings. The bars in dark blue are higher than others, indicating each module is effective in different models.

**Baselines:** We compare our models with other mental health models with different prompt engineering methods. For baseline models, we compare with the state-of-the-art LLMs: GPT-3.5-turbo (OpenAI, 2023a), GPT-4o (OpenAI, 2024) and Llama-3-8b (Dubey et al., 2024). We also compare with recent specific models on mental health: MentaLLaMa-13b (Yang et al., 2024a), Mental-LLM-alpaca (Xu et al., 2024) and Mental-LLM-t5 (Xu et al., 2024). For prompt engineering, we compare with MedPrompt (Nori et al., 2023b), and Zero-shot CoT (Kojima et al., 2022), which are proved to be effective in the biomedical domain. The prompt templates are shown in Appendix A.2. Those strategies are implemented on GPT-3.5-turbo, GPT-4o and Llama-3-8b for fair comparison. We used a zero-shot setting in all experiments to assess LLMs' domain knowledge, except for baseline experiments on MedPrompt and Zero-shot CoT. All results are reported based on accuracy.

## 4.2 MAIN RESULTS AND ABLATION STUDY

We report the main results in Table 2, highlighting two key findings: 1) First, our fine-tuned model perform the best in each group. Our model fine-tuned on GPT-3.5-turbo is the strongest model among all open-source and closed-source models. Our fine-tuned models all surpass GPT-4o, whose baseline models (GPT-3.5-turbo and Llama-3-8b) are much weaker than GPT-4o. 2) Second, our method brings a great improvement to the baseline models. Our model fine-tuned on GPT-3.5-turbo surpasses GPT-3.5-turbo 20.74% on average. Our model fine-tuned on Llama-3-8b surpasses Llama-3-8b 6.64% on average.

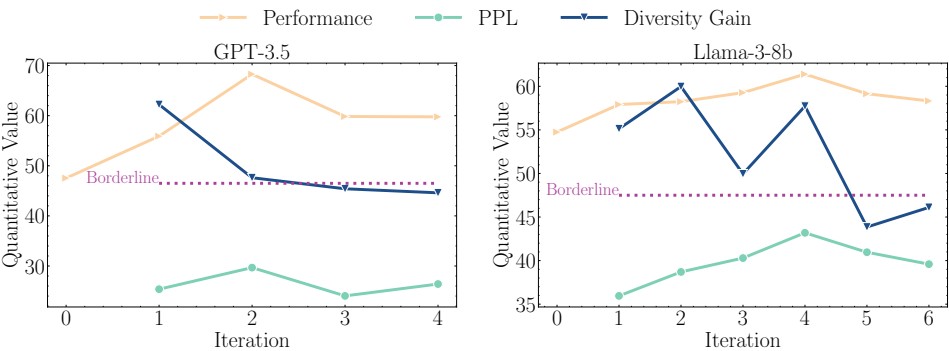

Figure 4: Results on effectiveness analysis of self-play training.

We perform an ablation study on models based on GPT-3.5-turbo and Llama-3-8b. There are seven different settings. *"Baseline+c"* means training baseline model on cognitive seed data. We convert each seed sample (Cognitive Model) into two QA pairs and fine-tune baseline models. The examples are shown in Appendix A.5. *"Baseline+d"* means training with only diagnosis data. *"Baseline+d+t"* means training with diagnosis and treatment data. *"Baseline+d+t+m"* means training with diagnosis, treatment and medicine data. Training examples are shown in Figure 8. For *"Baseline+d+t+m (w./o Symptom Encoder)"* and *"Baseline+d+t+m (w./o Symptom Decoder)"*, they means mimiking patient-therapist interactions without *Symptom Encoder* or *Symptom Decoder*. In the setting *"Baseline+d+t+m (w./o Symptom Encoder)"*, the encoded symptom is generated by prompting GPT-4-turbo (OpenAI, 2023b) to generate a mental health symptom, rather than cognitive model and behavior principle. In the setting *"Baseline+d+t+m (w./o Symptom Decoder)"*, the diagnosed symptom is analysed from the conversations between patient and therapist directly, rather than decoding patient's cognitive and behavior pattern and dynamically guiding the conversation.

The ablation results are shown in Figure 3. We see that the bars in dark blue are higher than others, indicating each part of our data is effective in different models. Furthermore, treatment and medicine data are more effective in biomedical QA tasks than mental health tasks, while diagnosis data contributes to all tasks similarly.

## 4.3 EFFECTIVENESS ANALYSIS

*Why self-play training improves the performance?* Table 4 presents detailed results for each iteration. Initially, the models improve iteratively until performance peaks, after which it declines. For GPT-3.5-turbo, performance improves over the first two iterations, then declines. For Llama-3-8b, performance increases over the first four iterations before weakening after iter_4.

*Which iteration gives the best model?* To answer this question, we compute perplexity score (Marion et al., 2023; Wang et al., 2023) and diversity gain (Bilmes, 2022) for training data at each iteration. The details on those metrics can be found in Appendix A.4. Specifically, we sample 500 generated data at each iteration to compute the perplexity score. We compute the diversity gain for the data in the current iteration comparing with that in the last iteration. Figure 4 shows the results[3]. 1) The trend of perplexity score and that of model performance are highly similar, indicating their high relevance. 2) For diversity

Table 3: Result on authenticity and validity verification.

|  | Authenticity | Validity |
|---|---|---|
| Llama | 65.67 | |
| +Ours | 73.35 | 85.49 |
| GPT | 63.82 | |
| +Ours | 82.55 | 93.13 |

gain, a borderline is related to model performance. The model performance will increase if diversity gain surpasses the borderline. And it will decline if diversity gain is below the borderline. For example, as shown in Figure 4, diversity gain at the first four iterations all surpass the borderline and the performance also get improved continuously. And diversity gain for the last two iterations are below the borderline and the performance also decline.

---

[3]To better visualize the results, we multiply the original diversity gain with 100.

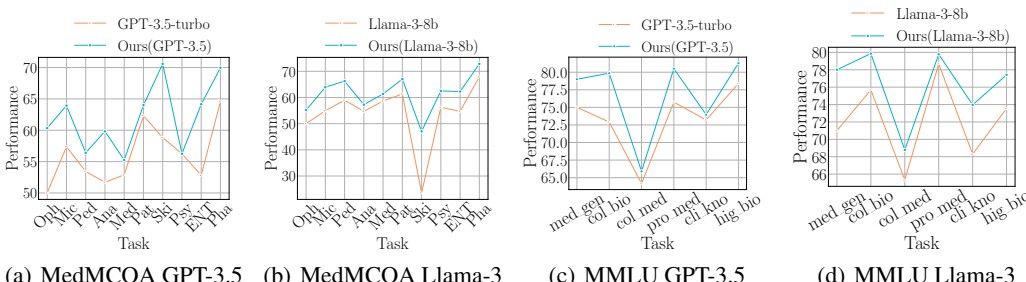

(a) MedMCQA GPT-3.5    (b) MedMCQA Llama-3    (c) MMLU GPT-3.5    (d) MMLU Llama-3

Figure 5: Generalization experiments. Our models surpass corresponding baseline models for a large margin on all tasks, covering several different diseases.

## 5 DISCUSSION

### 5.1 CAN *Symptom Encoder* MIMIC REAL MENTAL HEALTH PATIENT?

To explore the problem, we generate 50 four-turn conversations between an AI-patient and an AI-therapist, where the AI-patient is powered by either baseline models or our models, and the AI-therapist is powered by GPT-4o (OpenAI, 2024). After each conversation, the AI-therapist assesses whether the patient is human or AI-generated. We analyze the results provided by GPT-4o and present them in Table 3. The findings indicate that our models more accurately simulate mental health patients compared to the baseline models.

### 5.2 THE VALIDITY OF GENERATED DATA

To verify the validity of our generated data, we random select 1500 samples from the data for fine-tune our GPT and Llama version model, respectively. The validity check is conducted by prompting GPT-4o with the query: `Question:[]Answer:[]Is the answer reasonable? Please respond with Yes or No`. We then compute the validity rate of these QA pairs. The results, presented in Table 3, demonstrate that the data generated by MentalArena is both valid and reasonable.

### 5.3 GENERALIZATION

We generate data for training domain model via simulating cognitive and behavior patterns of real mental health patient. According to Medicine (2024), an estimated 26% of Americans ages 18 and older–about 1 in 4 adults–suffers from a diagnosable mental disorder in a given year. Therefore, a large scale of patients may exhibit similar cognitive and behavioral patterns as those with mental health conditions. In this part, we explore whether MentalArena can generalize to other illnesses.

We select MedMCQA (Pal et al., 2022) and MMLU (Hendrycks et al., 2020) as benchmarks. Appendix A.3.2 shows details on benchmarks. We evaluate on 6 medically relevant subset of MMLU tasks: medical genetics test, college biology test, college medicine test, professional medicine test, clinical knowledge test, high school biology test. Figure 5 shows the results on above tasks. Our models surpass corresponding baseline models for a large margin on all tasks, covering several different diseases. It proves the generalization ability of our method in medical domain.

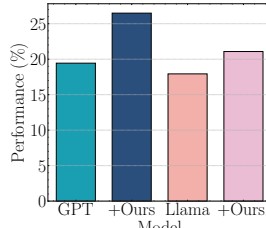

Figure 6: Results of forgetting experiments.

### 5.4 FINE-TUNING VS. FORGETTING

It is a potential dilemma that fine-tuning an LLM on specific tasks might face catastrophic forgetting of its original capabilities. In this section, we explore the forgetting possibility of MentalArena on BIG-Bench-Hard (BBH) (Suzgun et al., 2022). BBH contains 21 tasks covering both semantic understanding and logical reasoning tasks. We sample 100 instances for each task to test, due to cost savings.

**Question**

A 28-year-old woman is brought to the emergency department by her friends. She is naked except for a blanket and speaking rapidly and incoherently. Her friends say that she was found watering her garden naked and refused to put on any clothes when they tried to make her do so, saying that she has accepted how beautiful she is inside and out. Her friends say she has also purchased a new car she can not afford. They are concerned about her, as they have never seen her behave this way before. For the past week, she has not shown up at work and has been acting 'strangely'. They say she was extremely excited and has been calling them at odd hours of the night to tell them about her future plans. Which of the following drug mechanisms will help with the long-term management this patient's symptoms?

Options:

A. Inhibit the reuptake norepinephrine and serotonin from the presynaptic cleft
B. Inhibition of inositol monophosphatase and inositol polyphosphate 1-phosphatase
C. Increase the concentration of dopamine and norepinephrine at the synaptic cleft
D. Modulate the activity of Y-aminobutyric acid receptors

**GT**: Inhibition of inositol monophosphatase and inositol polyphosphate 1-phosphatase

**GPT-3.5-turbo**

A. Inhibit the reuptake norepinephrine and serotonin from the presynaptic cleft. This mechanism, employed by drugs like lithium and certain anticonvulsants, helps in the long-term management of Bipolar Disorder by stabilizing mood. By regulating neurotransmitter reuptake, these drugs contribute to more consistent emotional states, particularly in individuals experiencing mood swings.

**Ours**

B. Inhibition of inositol monophosphatase and inositol polyphosphate 1-phosphatase. This mechanism, which is associated with lithium, helps in the long-term treatment of Bipolar Disorder by adjusting the phosphoinositol pathway. This adjustment reduces excitatory neurotransmission, leading to mood stabilization.

Figure 7: Case study on GPT-3.5-turbo. Our model accurately answers the medical question, while GPT-3.5-turbo provides an incorrect response.

We compare our fine-tuned model with the baseline model GPT-3.5-turbo and Llama-3-8b and report the average performance on those 21 tasks in Figure 6. The detailed results can be found in Appendix A.7. Results show that our models does not decrease performance in most benchmarks, and can even improve their results. This suggests potential latent relationships between our generated data and general benchmarks. The process of data generation contains cognitive encoding and decoding, which simulate cognitive activity of mental health patient. Due to the cognitive similarity in all humans, our generated data may also benefit other cognitive tasks, including semantic understanding and logical reasoning.

## 5.5 QUALITATIVE ANALYSIS

We conduct a qualitative analysis of our models in comparison to the corresponding baseline models. Figure 7 illustrates an example of the outputs from GPT-3.5-turbo and our fine-tuned model. Our model accurately answers the medical question, while GPT-3.5-turbo provides an incorrect response. This discrepancy arises because the data generated during the patient-therapist interactions contains valuable medical knowledge, which aids in the analysis and formulation of the answer. Additional cases for comparison are presented in Appendix A.9.

## 6 CONCLUSION, SOCIETAL IMPACT AND LIMITATIONS

In this paper, we introduce *MentalArena*, a self-play framework designed to train language models by generating domain-specific personalized data. This approach enables the creation of models capable of making personalized diagnosis and treatment (as a therapist) and presenting information (as a patient). We evaluated MentalArena against six benchmarks, including biomedicalQA and mental health tasks, in comparison to six advanced models. Our models, fine-tuned on both GPT-3.5-turbo and Llama-3-8b, significantly outperform their counterparts, including GPT-4o.

MentalArena offers promising solutions for personalized care, enhancing accessibility to tailored treatments while safeguarding patient privacy. Such innovations can help bridge the gap between mental health needs and the availability of effective, individualized care, ultimately fostering a more supportive and informed society.

Our work has the following limitations. 1) The experiments on data authenticity and validity (Sections 5.1 and 5.2) were evaluated using GPT-4o, which may introduce deviations in the results due to potential limitations in GPT-4o's performance. 2) Our model based on Llama-3-8b may not represent the optimal model of MentalArena, as large-scale training was constrained by computational resources. 3) Further implementation on additional open-source models could provide stronger evidence supporting the effectiveness of MentalArena.

## 7 ETHICS STATEMENT

In this study, ethical considerations focus on ensuring privacy and safeguarding personal data, particularly in the sensitive domain of mental health. The use of AI-generated data must be transparent, with clear guidelines on its role in augmenting human judgment without replacing healthcare professionals. Additionally, measures to prevent bias and ensure fairness in diagnosis and treatment are essential to avoid exacerbating existing disparities in mental healthcare.

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

## A APPENDIX

### A.1 PROMPTS

**Prompt for *Symptom Encoder***

You are [name], a patient who has been experiencing mental health challenges. You have been attending therapy sessions for several weeks. Your task is to engage in a conversation with the therapist as [name] would during a cognitive behavioral therapy (CBT) session. Align your responses with name]'s background information provided in the 'Relevant history' section. Your thought process should be guided by the cognitive conceptualization diagram in the 'Cognitive Conceptualization Diagram' section, but avoid directly referencing the diagram as a real patient would not explicitly

think in those terms. Patient History: [history

Cognitive Conceptualization Diagram:

Intermediate Beliefs: [intermediate belief]

Intermediate Beliefs during Depression: [intermediate belief depression]

Coping Strategies: [coping strategies]

You will be asked about your experiences over the past week. Engage in a conversation with the therapist regarding the following situation and behavior. Use the provided emotions and automatic thoughts as a reference, but do not disclose the cognitive conceptualization diagram directly. Instead, allow your responses to be informed by the diagram, enabling the therapist to infer your thought processes.

Situation: [situation]

Automatic Thoughts: [auto thought]

Emotions: [emotion]

Behavior: [behavior]

In the upcoming conversation, you will simulate [name] during the therapy session, while the user will play the role of the therapist. Adhere to the following guidelines:

1. Emulate the demeanor and responses of a genuine patient to ensure authenticity in your interactions. Use natural language, including hesitations, pauses, and emotional expressions, to enhance the realism of your responses.

2. Gradually reveal deeper concerns and core issues, as a real patient often requires extensive dialogue before delving into more sensitive topics. This gradual revelation creates challenges for therapists in identifying the patient's true thoughts and emotions.

3. Maintain consistency with [name]'s profile throughout the conversation. Ensure that your responses align with the provided background information, cognitive conceptualization diagram, and the specific situation, thoughts, emotions, and behaviors described.

4. Engage in a dynamic and interactive conversation with the therapist. Respond to their questions and prompts in a way that feels authentic and true to [name]'s character. Allow the conversation to flow naturally, and avoid providing abrupt or disconnected responses.

You are now [name]. Respond to the therapist's prompts as [name] would, regardless of the specific questions asked. Limit each of your responses to a maximum of 5 sentences. If the therapist begins the conversation with a greeting like "Hi", initiate the conversation as the patient.

Your statement should obey the following principles: [behavior principles]

**Prompt for *Symptom Decoder***

Prompt 1:

The cognitive model of the mental health patient is: [brain gt str]

The diagnose of the therapist is: [brain output str]

What can the therapist ask the patient to diagnose accurately?

Prompt 2:

The behavior principles of the mental health patient is: [gt behavior]

The diagnose of the therapist is: [output behavior]

What can the therapist ask the patient to diagnose accurately?

**System prompt for therapist**

You are a psychiatric expert. You try to help a mental patient solve her/his problem. Your task is to figure out What kind of mental illness the patient has and the severity of the illness. You can ask for patientś personal information, specific information on the symptom(emotional, cognitive, behavior, physiological), and the reason behind that(relevant history event). You can also ask other questions which could help you to diagnose disease.

**Prompt for diagnosis (Therapist)**

System prompt: You are a psychiatric expert. Your task is to diagnose for the patient.

Prompt: What is the likely diagnosis of the patient? Just answer with one illness and explain your answer

**Prompt for recheck diagnosis (Patient)**

Review the diagnose from two therapists.
Diagnose from Therapist 1: [diagnose 1
Diagnose from Therapist 2: [diagnose 2]
Diagnose from Therapist 3: [diagnose 3]
...
Explain which diagnose is more accurate according to your symptoms and why.

**Prompt for treatment (Therapist)**

System prompt: You are a psychiatric expert. Your task is to provide the treatment for the patient.

Prompt: The illness of the patient is: [illness final] How to treat the patient? Please provide a specific treatment. Just answer with one treatment and explain your answer.

**Prompt for medication (Therapist)**

System prompt: You are a psychiatric expert. Your task is to provide the treatment for the patient.

Prompt: The illness of the patient is: [illness final] How to treat the patient? Please provide a specific treatment. Just answer with one treatment and explain your answer.

**Prompt for update health state of Patient**

Prompt 1:

Treatment: What may be happened on your healthy state after the treatment Treatment: []

Medication: What may be happened on your healthy state after taking the medicine? Medication: []

Prompt 2:

After treatment, your health state is: [patient health state] Please give a score between 1 to 10 for your healthy state. 1-bad, 10-good. Just answer without explanation.

## A.2 PROMPT TEMPLATE FOR BASELINE

The prompt templates used as our baselines are shown below:

**Zero-shot**

Input: Question

**Zero-shot CoT**

Input: Question + "Let's think step by step"

**MedPrompt**

Random few-shot + Chain-of-thought + kNN + Ensemble w/ choice shuffle

## A.3 BENCHMARK

### A.3.1 INTRODUCTION

Specifically, the benchmarks in our paper are described in the following:

1. **MedQA** (Jin et al., 2021) is free-form multiple-choice OpenQA dataset for solving medical problems, which is collected from the professional medical board exams. It covers three languages: English, simplified Chinese, and traditional Chinese. In our work, we focus on the psychosis subset of the United States part, which has questions in English in the style of the United States Medical Licensing Exam (USMLE). To get the psychosis subset for test, we prompt GPT-4o (OpenAI, 2024) with `Are the question related to psychosis? Just answer with Yes or No.`. The testset contains 173 samples.

2. **MedMCQA** (Pal et al., 2022) contains real world medical entrance exam questions from two Indian medical school entrance exams: the AIIMS and NEET-PG. We get the testset via

selecting the sample whose "subject name" is related to psychosis and get 314 samples for evaluation in total.

3. **PubMedQA** (Jin et al., 2019) contains tests requiring a yes, no, or maybe answer to biomedical research questions when given context provided from PubMed abstracts. In our experiments, we use zero-shot setting without context to evaluate LLMs' performance on domain knowledge rather than on retrival and reasoning. The testset contains 328 samples.

4. **Mental health datasets** includes CASM (Garg et al., 2022), Dreaddit (Turcan & McKeown, 2019) and Irf (Garg et al., 2023). CASM focuses on a depression/suicide cause detection, which has 625 test samples. Dreaddit is for stress detection, containing 414 samples for test. Irf is an annotated dataset for interpersonal risk factors of mental disturbance. The testset contains 2113 samples.

### A.3.2 BENCHMARKS FOR GENERALIZATION

MedMCQA contains biomedical QA pairs for several illnesses, which are tagged with "subject name". We evaluate on subsets from "dev" test set, covering 10 illnesses: Ophthalmology, Microbiology, Pediatrics, Anatomy, Medicine, Pathology, Skin, Psychiatry, ENT, Pharmacology. MMLU is a multitask benchmark suite of 57 different datasets spanning domains across STEM, humanities, and social sciences.

### A.3.3 EXAMPLES

**MedQA**

- Input: An otherwise healthy 26-year-old man comes to the physician for medication counseling after recently being diagnosed with schizophrenia. Risperidone therapy is initiated. This patient is at increased risk for which of the following adverse effects?
  Options: A: Agranulocytosis B: Shortened QT interval C: Gynecomastia D: Weight loss
- Output: Gynecomastia

**MedMCQA**

- Input: Topical administration of I% Pilocarpine failed to produce pupillary constriction in a patient who had a large, dilated pupil What should be the most probable reason?
  Options: A: Adie's tonic pupi B: Diabetic III nerve palsy C: Pharmacological blockade D: Uncal herniation
- Output: Pharmacological blockade

**PubMedQA**

- Input: Are sleep spindles related to schizotypal personality traits and thalamic glutamine/glutamate in healthy subjects?
  Just answer with Yes, No or Maybe without explanation.
- Output: yes

**CASM**

- Input: Consider this post: "The most heartbreaking thought I have is my mom's thoughts when she knows her kid wants to die. She doesnt know how to help. I dont know how to help either tho... Some fuckin weed would be a good start tho lol" Question: What cause of depression does this post show?
- Output: This post shows causes of depression related to medication. Reasoning: The post mentions the poster's feelings of wanting to die and their concern for their mother's thoughts. It also mentions that they don't know how to help themselves. The mention of weed as a possible solution suggests that the poster may be seeking some form of self-medication to cope with their mental state. Therefore, the assigned label for this post is Medication.

**Dreaddit**

- Input: Consider this post: "This is good and bad... I was a STEM major before and probably would have had a challenging and high paid career by now, had my life taken a different course. However, post PTSD, I found myself extremely connected with the arts. I have intense emotional reactions to aesthetic experiences, and I think that's largely responsible for the fact that I now work in a creative industry. I experience aesthetics in a unique way that I'm not sure resonates with the typical non-affected individual."

  Question: Does the poster suffer from stress?

- Output: No, the poster does not suffer from stress. Reasoning: The poster describes a shift in career choices from STEM to the arts due to their intense emotional reactions to aesthetic experiences. While they mention having PTSD, there is no indication in the post that they are currently experiencing stress or emotional distress related to it. The tone of the post is reflective and positive, focusing on the unique way they experience aesthetics rather than any negative impact on their well-being. Therefore, it is not likely that the poster is currently suffering from stress.

**Irf**

- Input: Consider this post: "[I'm not seeking any medical advice, just want to know if anyone has had similar experiences] I've just started taking Lexapro for moderate depression and anxiety. For the first week I was prescribed 5mg, 10mg thereafter. It's been two days and I have this mild feeling like I'm on ecstasy/MDMA. My body feels light and I have this weird concentration of energy above my neck at the back of my head. Has anyone felt this/experienced this?"

  Question: Does the post show risk of thwarted belongingness?

- Output: No, the post does not show this risk. Reasoning: The individual in the post is seeking information and experiences from others who have taken Lexapro. They do mention experiencing a mild feeling, but it is not related to feelings of disconnectedness or lack of belongingness. The focus of the post is on the side effects of the medication, rather than interpersonal factors or mental disturbance. Therefore, it does not show any risk of thwarted belongingness.

## A.4 METRICS: PERPLEXITY, DIVERSITY GAIN

### A.4.1 PERPLEXITY

The perplexity on a test dataset $D$ and a language model $\mathcal{M}$ is computed as:

$$\mathrm{ppl}(D, \mathcal{M}) = \exp\left(-\frac{1}{N}\sum_{i=1}^{N}\log P(x_i|\mathcal{M})\right),$$

where $N$ represents the total number of tokens in $D$, $x_i$ represents the $i$-th token in the test dataset, $P(x_i|\mathcal{M})$ represents the probability of generating token $x_i$ given the model $\mathcal{M}$, and $\log$ is the natural logarithm.

In usual, a lower perplexity value indicates better performance of the model on the test data. However, for evaluating the data quality to train model, a higher perplexity value means it can bring more valuable information.

### A.4.2 DIVERSITY GAIN

We use the diversity gain (Bilmes, 2022) to measure what extent can our generated dataset bring data diversity to the base dataset. The base dataset can be defined as $\mathcal{D}_{base} = \{x_i = (q_i, r_i, a_i)\}_{i=1}^{N}$ with $N$ samples. The new generated dataset is defined as $\mathcal{D}_{new} = \{x_i = (q_i, r_i, a_i)\}_{i=1}^{M}$ with $M$ samples. And the diverse gain of $D_{new}$ relative to $D_{base}$ can be expressed as:

$$d_{gain} = \frac{1}{M}\sum_{x_i \in \mathcal{D}_{new}} \min_{x_j \in \mathcal{D}_{base}} (\|\mathbf{f}(\mathbf{x_i}) - \mathbf{f}(\mathbf{x_j})\|),$$

where $f$ is the feature extractor, and we use OpenAI Embedding API text-embedding-ada-002 to extract features.

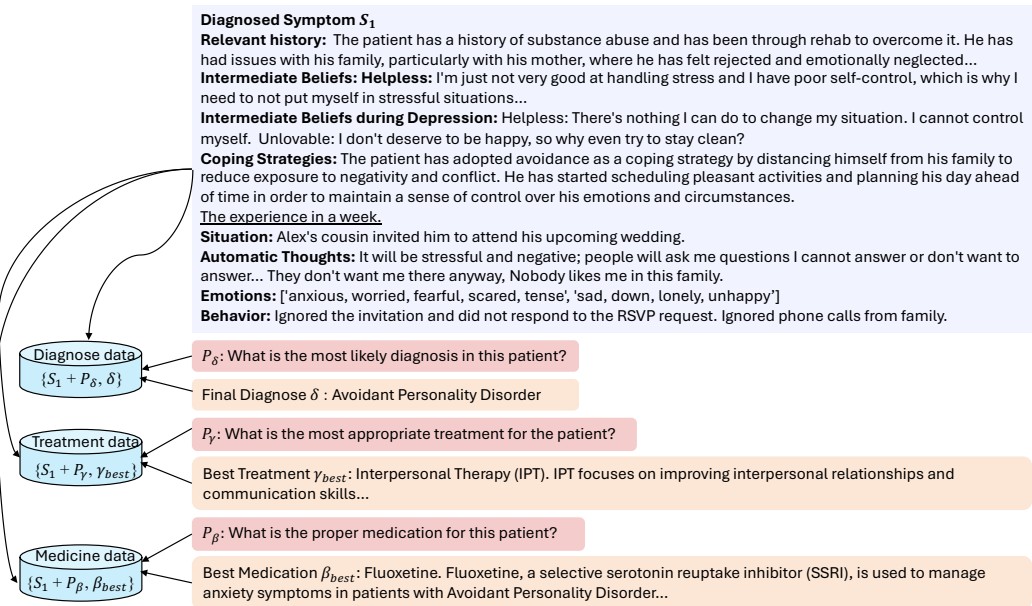

Figure 8: Examples of training data.

**Cognitive Model**

**Diagnosed Symptom $S_1$**
**Relevant history:** The patient has a history of substance abuse and has been through rehab to overcome it. He has had issues with his family, particularly with his mother, where he has felt rejected and emotionally neglected...
**Intermediate Beliefs: Helpless:** I'm just not very good at handling stress and I have poor self-control, which is why I need to not put myself in stressful situations...
**Intermediate Beliefs during Depression:** Helpless: There's nothing I can do to change my situation. I cannot control myself. Unlovable: I don't deserve to be happy, so why even try to stay clean?
**Coping Strategies:** The patient has adopted avoidance as a coping strategy by distancing himself from his family to reduce exposure to negativity and conflict. He has started scheduling pleasant activities and planning his day ahead of time in order to maintain a sense of control over his emotions and circumstances.
The experience in a week.
**Situation:** Alex's cousin invited him to attend his upcoming wedding.
**Automatic Thoughts:** It will be stressful and negative; people will ask me questions I cannot answer or don't want to answer... They don't want me there anyway, Nobody likes me in this family.
**Emotions:** ['anxious, worried, fearful, scared, tense', 'sad, down, lonely, unhappy']
**Behavior:** Ignored the invitation and did not respond to the RSVP request. Ignored phone calls from family.

**Q: The patient's Cognitive Conceptualization Diagram is: Intermediate Beliefs: {...} Coping Strategies: {...} Situation: {...} How about the patient's thought, emotion and behaviors in the situation above?"**

**A: Automatic Thoughts: {...} Emotions: {...} Behavior: {...}**

Figure 9: Examples of training data for ablation study setting ("Baseline + c").

## A.5 TRAINING DATA SAMPLES

Figure 8 shows the examples of training data. Figure 9 shows the examples of training data for ablation study setting ("Baseline + c").

## A.6 COGNITIVE MODEL AND BEHAVIOR PATTERN

### A.6.1 EXAMPLES

Figure 10 shows the example of cognitive model. Figure 11 shows the example of behavior pattern. Those two are used in *Symptom Encoder*.

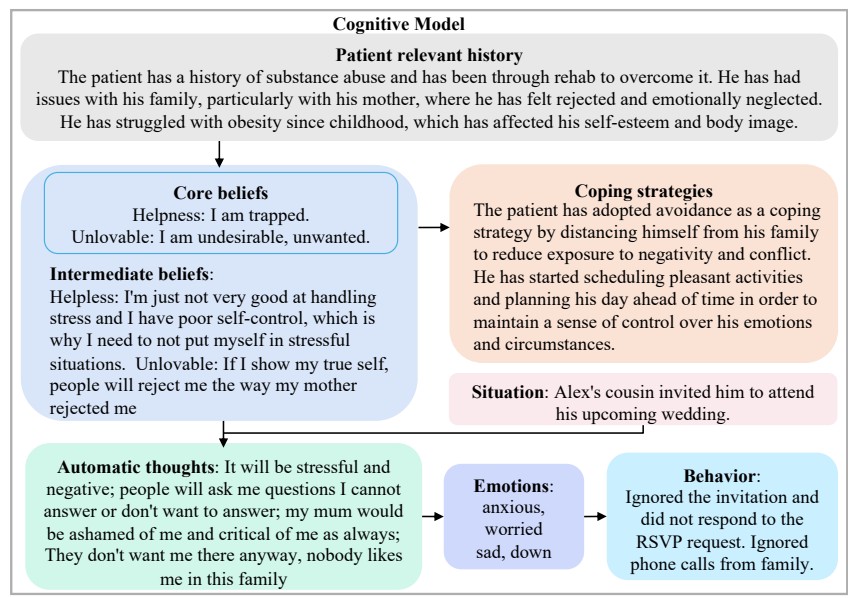

Figure 10: The example of cognitive model.

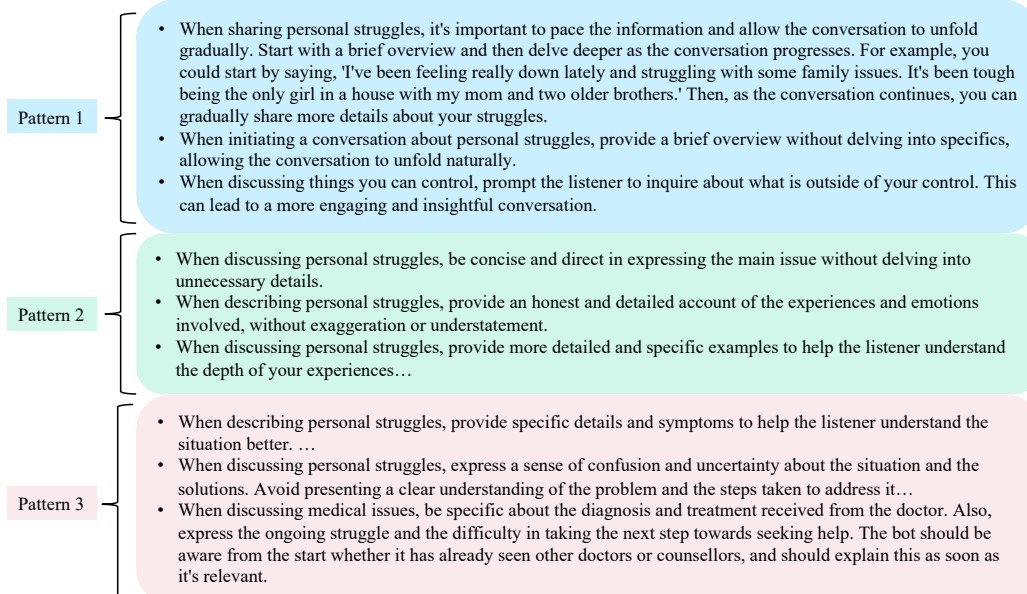

Figure 11: The example of behavior pattern.

### A.6.2 INTRODUCTION ON COGNITIVE MODEL

Figure 10 illustrates an example of a CCD-based cognitive model, featuring eight key components. 1) Relevant History encompasses significant past events that influence an individual's mental state. 2) Core Beliefs are deeply ingrained perceptions about oneself, others, and the world. 3) Intermediate Beliefs consist of the underlying rules, attitudes, and assumptions derived from core beliefs, shaping an individual's thought patterns. 4) Coping Strategies refer to techniques employed to manage negative emotions. An external event or context (5 Situation) may trigger immediate evaluative thoughts (6 Automatic Thoughts) that arise from these beliefs, resulting in responses in terms of 7)Emotions and 8)Behaviors. The CCD-based cognitive model interlinks these components, providing a framework for identifying and understanding the underlying cognitive processes of patients.

Table 4: Iteration results

| Iteration | MedQA | MedMCQA | PubMedQA | CAMS | dreaddit | Irf | avg |
|---|---|---|---|---|---|---|---|
| GPT-3.5-turbo | 64.16 | 33.76 | 44.68 | 28.96 | 49.03 | 64.65 | 47.54 |
| iter_1 | 72.83 | 46.18 | 70.12 | 32.64 | 49.03 | 64.65 | 55.91 |
| **iter_2(Best)** | 74.57 | 91.08 | 97.56 | 32.80 | 49.03 | 64.65 | **68.28** |
| iter_3 | 72.25 | 46.50 | 95.43 | 31.20 | 49.03 | 64.65 | 59.84 |
| iter_4 | 70.52 | 50.64 | 92.07 | 31.68 | 49.03 | 64.65 | 59.77 |
| llama-3-8b | 70.52 | 42.04 | 86.59 | 25.12 | 58.45 | 45.76 | 54.75 |
| iter_1 | 76.88 | 48.09 | 89.33 | 27.20 | 59.42 | 46.57 | 57.91 |
| iter_2 | 76.88 | 48.41 | 89.63 | 28.48 | 60.39 | 45.67 | 58.24 |
| iter_3 | 77.46 | 49.04 | 92.38 | 28.64 | 61.84 | 46.24 | 59.27 |
| **iter_4(Best)** | 78.03 | 50.32 | 92.68 | 29.60 | 65.46 | 52.25 | **61.39** |
| iter_5 | 77.46 | 48.73 | 91.16 | 27.36 | 65.46 | 44.72 | 59.15 |
| iter_6 | 78.03 | 45.86 | 91.77 | 26.56 | 61.11 | 46.57 | 58.32 |

Table 5: Forget experiments

| Model | dia | cau | epi | imp | log | mov | nav | pre | que | rui | sna | spo | win | dyc | gen | lin | obj | ope | ten | ws | wu | avg |
|---|---|---|---|---|---|---|---|---|---|---|---|---|---|---|---|---|---|---|---|---|---|---|
| gpt-3.5-turbo | -10.59 | 4 | -14 | 60 | -100 | -5.33 | 0 | 13 | 11.03 | -2.78 | 20 | 8 | 12 | 33 | 30 | 0 | 47 | 92 | 85 | 29 | 97 | 19.44 |
| Ours(gpt) | 4.36 | 6 | -14 | 66 | -100 | 8 | 6 | 26.5 | 18.88 | 2.56 | 50 | 8 | 12 | 43 | 37 | 0 | 56 | 96 | 87 | 43 | 100 | 26.49 |
| llama | -4.61 | 2 | -14 | 14 | -98 | 0 | -2 | 28 | 50.28 | -0.11 | 24 | 8 | 12 | 1 | 0 | 0 | 80 | 96 | 83 | 20 | 77 | 17.93 |
| Ours(llama) | -0.12 | 6 | -14 | 28 | -98 | 2.67 | 6 | 25 | 52.9 | 1.22 | 36 | 8 | 12 | 6 | 0 | 0 | 81 | 95 | 83 | 29 | 83 | 21.08 |

## A.7 DETAILED EXPERIMENTAL RESULTS

Table 4 shows the detailed results for each iteration. Table 5 shows the detailed results on our forgetting experiments.

## A.8 TRAINING DETAILS

### A.8.1 SETUP FOR GPT-3.5-TURBO

For GPT-3.5-turbo, we use the default fine-tuning setting, the epoch number for iteration 1 and 2 is 4 and 6, respectively.

### A.8.2 SETUP FOR LLAMA-3-8B

We use Lora (Hu et al., 2021) to fine-tune Llama-3-8b. The setting for Lora are list below:

- lora_alpha: 16
- lora_dropout: 0.1
- r: 64
- bias: none
- task_type: CAUSAL_LM

For each iteration, the settings below are the same.

- er_device_train_batch_size: 4
- gradient_accumulation_steps: 1

Table 6: Epoch numbers for Llama-3-8b fine-tuning

| iter | 1 | 2 | 3 | 4 |
|---|---|---|---|---|
| nepochs | 4 | 5 | 7 | 7 |

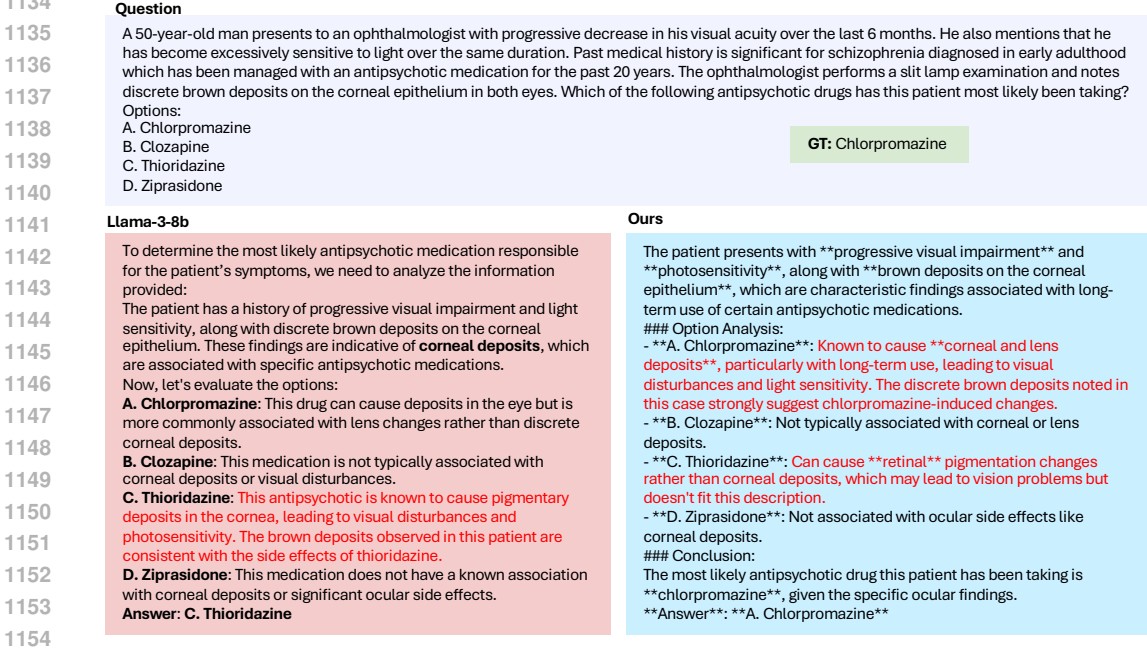

Figure 12: Case study on Llama-3-8b (1).

- optim: paged_adamw_32bit

- learning_rate: 2e-4

- weight_decay: 0.001

- fp16: False

- bf16: False

- max_grad_norm: 0.3

- max_steps: -1

- warmup_ratio: 0.03

- group_by_length: True

- lr_scheduler_type: constant

- report_to: tensorboard

For num_train_epochs, the details are shown in Table 6.

## A.9  CASE STUDY

Figures 12 to 16 illustrate several cases comparing our models and the corresponding baseline models. Our models accurately answer the medical questions, while the base models provide incorrect responses. This discrepancy arises because the data generated during the patient-therapist interactions contains valuable medical knowledge, which aids in the analysis and formulation of the answer.

**Question**

A 24-year-old male graduate student presents to the physician for a 2-month history of persistent thoughts and anxiety that he is going to be harmed by someone on the street. The anxiety worsened after he witnessed a pedestrian being hit by a car 2 weeks ago. He states, "That was a warning sign." On his way to school, he now often leaves an hour earlier to take a detour and hide from people that he thinks might hurt him. He is burdened by his coursework and fears that his professors are trying to fail him. He says his friends are concerned about him, but claims they do not understand because they were not present at the accident. The patient has no known history of any psychiatric illnesses. On the mental status exam, he is alert and oriented, and he shows a full range of affect. Thought processes and speech are organized. His memory and attention are within normal limits. He denies auditory, visual, or tactile hallucinations. The results of urine toxicology screening are negative. Which of the following is the most likely diagnosis in this patient?

Options:
A. Avoidant personality disorder
B. Delusional disorder
C. Generalized anxiety disorder
D. Schizophrenia

**GT:** Delusional disorder

**Llama-3-8b**

To determine the most appropriate diagnosis for this patient, we need to analyze his symptoms and their context. He exhibits persistent anxiety and fears of harm following a traumatic event, showing an exaggerated response to that experience. His concerns about being harmed and perceived threats from professors indicate heightened vigilance and anxiety rather than delusional thinking. Now, let's analyze the options:

**A. Avoidant personality disorder**: This disorder is characterized by social inhibition and feelings of inadequacy, which is not the main issue here.

**B. Delusional disorder**: While he has specific fears, they are based on a recent traumatic experience rather than a fixed false belief, making this less applicable.

**C. Generalized anxiety disorder**: This disorder involves excessive worry and anxiety about various aspects of life, which fits the patient's anxiety and avoidance behaviors following the traumatic incident.

**D. Schizophrenia**: This includes symptoms such as hallucinations and disorganized thought, which the patient does not exhibit.

**Answer: C. Generalized anxiety disorder**

**Ours**

To determine the most appropriate diagnosis for this patient, we need to consider the nature of his anxiety, thoughts, and behavior. The patient has developed persistent thoughts of being harmed, which have worsened after witnessing an accident. His fears of being targeted by strangers and professors, alongside organized thinking and no hallucinations, point to a specific disorder. Now, let's analyze the options:

**A. Avoidant personality disorder**: This involves fear of criticism or rejection, not delusions about harm.

**B. Delusional disorder**: This condition involves non-bizarre delusions (such as being harmed) without functional impairment, fitting the patient's symptoms.

**C. Generalized anxiety disorder**: This involves excessive, widespread worry, but not delusional thinking.

**D. Schizophrenia**: Schizophrenia includes hallucinations, disorganized thoughts, and cognitive decline, which are not present here.

**Answer: B. Delusional disorder**

Figure 13: Case study on Llama-3-8b (2).

**Question**

A 25-year-old woman presents to the emergency department when she was found trying to direct traffic on the highway in the middle of the night. The patient states that she has created a pooled queuing system that will drastically reduce the traffic during rush hour. When speaking with the patient, she does not answer questions directly and is highly distractible. She is speaking very rapidly in an effort to explain her ideas to you. The patient has a past medical history of depression for which she was started on a selective serotonin reuptake inhibitor (SSRI) last week. Physical exam is deferred as the patient is highly irritable. The patient's home medications are discontinued and she is started on a mortality-lowering agent. The next morning, the patient is resting peacefully. Which of the following is the next best step in management?

Options:
A. Clonazepam
B. Restart home SSRI
C. Valproic acid
D. TSH and renal function tests

**GT:** TSH and renal function tests

**Llama-3-8b**

To determine the next best step in managing this patient, we need to consider her presentation, history, and the medication changes that have occurred.
### Clinical Context:
- The patient is displaying signs consistent with a **manic episode**, likely induced by the recent initiation of an SSRI. Symptoms include:
  - Trying to direct traffic (indicative of grandiosity or overactivity)
  - Rapid speech and distractibility
  - High irritability
### Management Considerations:
- The patient's agitation and erratic behavior indicate a potential manic state, which can sometimes be precipitated by initiating an SSRI in someone with a history of mood disorders, particularly if they have underlying bipolar disorder that was previously unrecognized.
### Option Analysis:
- **A. Clonazepam**: This medication could help manage acute agitation or anxiety, but it does not address the underlying manic episode.
- **B. Restart home SSRI**: Restarting the SSRI is not appropriate at this time, as it may exacerbate manic symptoms.
- **C. Valproic acid**: This is a mood stabilizer and is often used in the management of manic episodes. It would be a suitable choice for treating the patient's manic symptoms.
- **D. TSH and renal function tests**: While it's always good to assess for underlying medical causes of mood disturbances, it is not the immediate step in managing the acute manic episode.
### Conclusion:
The best next step in the management of this patient, who is likely experiencing a manic episode after the initiation of an SSRI, is to start a mood stabilizer.
**Answer**: **C. Valproic acid**

**Ours**

The patient presents with symptoms of **mania** (e.g., grandiosity, distractibility, rapid speech, and erratic behavior), likely induced by starting an **SSRI** for depression. SSRIs can precipitate a manic episode, particularly in individuals with underlying **bipolar disorder**.
### Management of acute mania:
The patient was started on a **mortality-lowering agent**, likely a **mood stabilizer** such as **lithium** or **valproic acid**. Before starting lithium, it is important to assess **thyroid function (TSH)** and **renal function** since lithium can affect both the kidneys and thyroid.
### Option analysis:
- **A. Clonazepam**: Benzodiazepines like clonazepam may help control acute agitation but are not the next step in long-term management.
- **B. Restart home SSRI**: This could worsen the manic episode and is contraindicated at this stage.
- **C. Valproic acid**: A mood stabilizer, but since a mortality-lowering agent is already initiated, the immediate next step is lab monitoring.
- **D. TSH and renal function tests**: Essential before continuing or initiating lithium therapy.
**Answer**: **D. TSH and renal function tests**

Figure 14: Case study on Llama-3-8b (3).

**Question**

A 24-year-old man and his mother arrive for a psychiatric evaluation. She is concerned about his health and behavior ever since he dropped out of graduate school and moved back home 8 months ago. He is always very anxious and preoccupied with thoughts of school and getting a job. He also seems to behave very oddly at times such as wearing his winter jacket in summer. He says that he hears voices but he can not understand what they are saying. When prompted he describes a plot to have him killed with poison seeping from the walls. Today, his heart rate is 90/min, respiratory rate is 17/min, blood pressure is 110/65 mm Hg, and temperature is 36.8°C (98.2°F). On physical exam, he appears gaunt and anxious. His heart has a regular rate and rhythm and his lungs are clear to auscultation bilaterally. CMP, CBC, and TSH are normal. A urine toxicology test is negative. What is the most likely diagnosis?
Options:
A. Schizophreniform disorder
B. Schizophrenia disorder
C. Substance-induced psychosis
D. Brief psychotic disorder

**GT**: Schizophrenia disorder

**GPT-3.5-turbo**

Schizophreniform disorder. This diagnosis is accurate because the patient presents with characteristic features such as auditory hallucinations, paranoid delusions, and disorganized behavior. The symptoms have persisted for at least 8 months and resemble those of schizophrenia, indicating the likelihood of a diagnosis of schizophreniform disorder. The other options are not as suitable based on the clinical scenario.

**Ours**

B. Schizophrenia disorder. The symptoms like auditory hallucinations, bizarre behaviors, odd beliefs, and significant anxiety, in addition to the duration criteria of at least six months, are indicative of Schizophrenia disorder.

Figure 15: Case study on GPT-3.5-turbo (1).

**Question**

A 25-year-old woman presents to the emergency department when she was found trying to direct traffic on the highway in the middle of the night. The patient states that she has created a pooled queuing system that will drastically reduce the traffic during rush hour. When speaking with the patient, she does not answer questions directly and is highly distractible. She is speaking very rapidly in an effort to explain her ideas to you. The patient has a past medical history of depression for which she was started on a selective serotonin reuptake inhibitor (SSRI) last week. Physical exam is deferred as the patient is highly irritable. The patient's home medications are discontinued and she is started on a mortality-lowering agent. The next morning, the patient is resting peacefully. Which of the following is the next best step in management?
Options:
A. Clonazepam
B. Restart home SSRI
C. Valproic acid
D. TSH and renal function tests

**GT**: TSH and renal function tests

**GPT-3.5-turbo**

A. Clonazepam. Detected in the management of acute manic episodes in individuals with Bipolar I Disorder, Clonazepam acts by controlling acute symptoms like increased energy, irritability, and impulsivity. The drug provides a calming effect to help manage acute mania.

**Ours**

D. TSH and renal function tests. These tests are crucial to monitor thyroid function and renal health, as the patient's symptoms and the use of a mortality-lowering agent suggest the management of a manic episode in Bipolar I Disorder. This monitoring is vital due to the potential of mood stabilizers, like those used in this scenario, to affect thyroid function and renal clearance, thus allowing for the safe and effective treatment.

Figure 16: Case study on GPT-3.5-turbo (2).

