# OpenReview forum: "MentalArena: Self-play Training of Language Models for Diagnosis and Treatment of Mental Health Disorders"
_ICLR.cc/2025/Conference — ICLR 2025 Conference Withdrawn Submission_

### Official Review · Reviewer_KoRE · 2024-10-29

**Soundness:** 2
**Presentation:** 3
**Contribution:** 3
**Rating:** 6
**Confidence:** 4

**Summary:**

The paper introduces MentalArena, a self-play framework designed to train language models for the diagnosis and treatment of mental health disorders. The framework enables a language model to adopt both patient and therapist roles, generating synthetic data that simulates patient-therapist interactions. Through components such as the Symptom Encoder, Symptom Decoder, and Model Optimizer, MentalArena models cognitive and behavioral patterns associated with mental health patients and optimizes responses through iterative self-play. The authors demonstrate notable improvements in performance on benchmarks compared to existing models.

**Strengths:**

The use of self-play for mental health modeling is novel and aligns well with current challenges in healthcare data privacy. By generating synthetic data, the model mitigates the need for real-world mental health records, which are often inaccessible due to confidentiality concerns.

The authors conducted evaluations on multiple biomedical QA and mental health tasks, and the MentalArena model exhibited superior performance compared to various state-of-the-art language models.

 The integration of Symptom Encoder and Decoder modules helps in handling intent bias and enhances the model's realism by simulating human-like responses. This layered approach is thoughtful, tackling key challenges specific to mental health applications.

**Weaknesses:**

While synthetic data generation is advantageous for privacy, the paper could provide a deeper analysis of how well this data approximates real-world patient-therapist interactions. Evaluations involving feedback from human mental health professionals would strengthen claims of the model’s applicability.

Although the authors claim that MentalArena can generalize to other medical domains, the paper lacks detailed experiments beyond mental health, with limited benchmarks for validation. Future work could expand these evaluations to address potential generalization challenges.

The limitations section briefly mentions potential biases and computational constraints. However, ethical concerns, particularly the risk of misdiagnosis and over-reliance on automated tools in mental healthcare, are areas where a more extensive discussion would be valuable.

**Questions:**

See above

---

### Official Review · Reviewer_HFNq · 2024-10-31

**Soundness:** 3
**Presentation:** 3
**Contribution:** 3
**Rating:** 5
**Confidence:** 3

**Summary:**

In this paper, the authors propose a novel self-play tuning framework to enhance the ability of LLMs for diagnosing mental symptoms. This enhanced model can simultaneously act as both the patient and therapist. Through the support of cognitive models and behavioral patterns, the data can be disentangled to provide more effective diagnosis, treatment, and medication recommendations, thereby fine-tuning the LLM. Evaluation on diverse benchmarks demonstrates significant improvements with the self-play tuning approach, highlighting the advantages of this framework. However, several issues still need to be clarified and addressed. I will update my score if my concerns are adequately resolved.

**Strengths:**

* The concept of having the LLM act as both the patient and therapist is novel and intriguing.
* The contribution to mental health research is substantial.
* The paper includes a wide range of evaluation benchmarks, providing robust assessment.
* The proposed model, MentalArea, demonstrates superior performance across all benchmarks.

**Weaknesses:**

* The workflow of the entire self-play tuning process is unclear, you may update figure 1.
* There is a lack of illustration for the symptom encoder.
* The model inference process is not well-illustrated.
* A sensitivity analysis on the alignment threshold between the symptom encoder and decoder is missing.
* The selection of baseline models appears unfair. Since the base model used here is LLaMA-3-8b, why not either upgrade the base model of MentaLLaMa to LLaMA-3-8b or downgrade your model to LLaMA-13b for consistency?
* Related work could be improved by adding a discussion of similar research studies.

   [1] Chen, Z., Deng, Y., Yuan, H., Ji, K., & Gu, Q. Self-Play Fine-Tuning Converts Weak Language Models to Strong Language Models. In Forty-first International Conference on Machine Learning.

**Questions:**

* In Section 3, MENTALARENA, what is the base model used for the symptom encoder and decoder? If I understand correctly, both the encoder and decoder use either LLaMA-3 or GPT-3.5, correct? Figure 1 is somewhat unclear on this point.
* In Section 3, MENTALARENA, how is data handled if it never reaches the defined alignment threshold in the symptom decoder?
* In Section 3.4, THERAPIST: SYMPTOM DECODER, how is the “best” defined? Is it determined using ground-truth labels?
* In the experiments, specifically in Table 2, why is there no fine-tuning or at least few-shot tuning for the baseline models GPT-3.5 and LLaMA-3? If this were done, would there still be a significant improvement?
* Why isn’t F1 score included as an evaluation metric, as it may be more suitable than accuracy? The MentaLLaMa model also uses F1 as an evaluation metric.
* In Section 4.2, MAIN RESULTS AND ABLATION STUDY, could you explain why each result with LLaMA-3 shows only marginal improvement, or even no improvement, across all datasets compared to fine-tuning on GPT-3.5?

**Details Of Ethics Concerns:**

I am uncertain whether these datasets are permitted for use with LLMs, which may suffer a manual review.

---

### Official Review · Reviewer_cj5A · 2024-11-01

**Soundness:** 2
**Presentation:** 2
**Contribution:** 2
**Rating:** 3
**Confidence:** 4

**Summary:**

The paper introduces MentalArena, a self-play framework designed to train language models for mental health diagnosis and treatment. This approach enables the creation of high-quality, domain-specific data while addressing privacy concerns, which is critical in mental health care. The framework comprises three modules: the Symptom Encoder, which simulates a human-like mental health patient from both cognitive and behavioral perspectives; the Symptom Decoder, which addresses intent bias; and the Model Optimizer, which iterates and improves the model based on generated interactions. Using MentalArena, they produce 18,000 samples and train the model on this dataset. The models fine-tuned on GPT-3.5 and Llama-3-8b significantly outperformed the base models (GPT-3.5-turbo and Llama-3-8b) on benchmarks related to mental health and biomedical QA tasks.

**Strengths:**

- MentalArena effectively addresses privacy challenges by generating domain-specific data through self-play, enabling model improvement without compromising patient confidentiality.
- Models fine-tuned through MentalArena achieve significant performance gains compared to baselines, indicating an effective enhancement of diagnostic and therapeutic capabilities.
- The framework includes mechanisms to minimize intent bias during patient-therapist interactions, enhancing the model's reliability.
- Evaluations across six benchmarks show that MentalArena-trained models excel in both mental health-specific and general medical tasks, demonstrating robust performance and generalizability.
- By generating 18,000 training samples, the authors provide a valuable resource for future research and model training in mental health domains.

**Weaknesses:**

- There may be some inconsistencies in the overall framework description. I will organize questions below to clarify these aspects.
- The inner workings of the Symptom Encoder and Decoder modules are not fully detailed, making it difficult to assess how effectively the framework models complex human symptoms and simulated dialogue. This is briefly touched upon in Table 3 ("Authenticity") but lacks further elaboration. Additionally, it is unclear whether the framework accurately reflects changes in the patient’s status over time.
- The quality of the synthetic dataset is also challenging to assess fully. In Table 3 ("Validity"), this aspect is merely evaluated through a simple query format, which does not provide an in-depth analysis.
- Although the framework generates synthetic data for mental health patients, it shows only marginal improvement on mental health-specific test sets, while demonstrating significant gains in biomedical QA tasks.

**Questions:**

- In the "OVERVIEW OF THE FRAMEWORK" section (line 215), the authors state, "As treatment and medication plans are administered to the patient, their health state evolves, reflected in the sequential updates of encoded symptoms" over k rounds. However, within this same section, the framework is described as calculating the semantic similarity between the encoded symptom (S0) and the diagnosed symptom (Sd), continuing the conversation until this similarity score exceeds 0.9. Afterward, the patient’s symptoms are analyzed, and a diagnostic plan is developed, followed by k rounds of treatment/medication, which result in k new symptoms. This description does not align with the initial explanation of sequential updates in the middle of Figure 1, or with the description in Section 3.4 and Figure 2, causing considerable confusion. Could the authors clarify this inconsistency? A revised version of one of these figures could be helpful.
- In the supplementary materials, the prompt for the Symptom Decoder simply includes the ground-truth cognitive model, behavioral principles, and current diagnosis, followed by the question: "What can the therapist ask the patient to diagnose accurately?" This does not align with the explanation that the decoded system extracts the cognitive and behavioral principles understood by the therapist and compares their similarity to the ground truth. Could the authors provide a more detailed explanation of how the Symptom Decoder operates in practice?
- From my understanding, the model’s fine-tuning is based solely on QA samples derived from the patient profile and the diagnosis, treatment, and medication generated through self-play. Is there evidence that this fine-tuning improves dialogue simulation performance? The authors could strengthen this section by providing comparative examples of dialogues pre- and post-fine-tuning, or by including metrics specifically designed to evaluate dialogue quality.
- In Table 2, the results across different settings for GPT-3.5-turbo all show identical performance results for the Dreaddit and IRF test sets. Is this an error?
- In Figure 3, what is the difference between the left and right graphs?
- In Figure 4, what is the Borderline and why don’t you display the PPL and Diversity Gain score in Iteration 0?
- The description of MedPrompt in the supplementary materials seems incorrect. As I understand it, "Random Few-Shot" is an ablation experiment for MedPrompt and is not actually part of the MedPrompt system.

---

### Official Review · Reviewer_Xvfk · 2024-11-02

**Soundness:** 2
**Presentation:** 3
**Contribution:** 2
**Rating:** 3
**Confidence:** 2

**Summary:**

This paper proposed the self-playing framework called MentalArena to train language models through patients and therapists conversation. The trained model can be used for diagnosing mental health disorders or providing treatment. The evaluation of diagnosis and treatment is through the multiple choice of medical QA datasets.

**Strengths:**

- this paper provides a unique way to generate synthetic conversational data for mental health patients. Due to the privacy issue, it is extremely hard to share the original call center data or conversational data about mental health patients.
- it is a smart way of using self playing to accumulate training data and train language model to generate better capability of answering different medQA datasets.
- Authors have compared their approach with several baseline methods, in a variety of Med QA datasets, and try to generalize it to other QA datasets.

**Weaknesses:**

- using medQA as proxy for mental health diagnosis can be problematic. Diagnosing mental health diseases require to follow medical guidelines (for example, diagnosis depression has guidelines for different age groups: https://www.apa.org/depression-guideline). Diagnosis is multiperspective, conversation is only one data points, others, such as blood test, lab test, and survey (such as PHQ-9 survey for depression), and it also requires to have longitudinal datapoints, rather than just a few turns of conversations.
- using medQA as proxy for mental health treatment is a far reach. Choosing the right choice in multiple choice exam for medical students does not mean that chatbot can do treatment. Besides prescribing antidepressant medicines, CBT could be another options which requires to take 10 -15 sessions, and evaluate the clinical outputs. CBT is not a simple conversation, rather a protocol driven therapy. It is unclear that chatbot can be treated as therapists. Chatbot can be a co-pilot, but definitely cannot be treated as an independent therapist.

**Questions:**

- can you please explain the issues for self-play for training language models. Is the self-playing version of conversation short, or stuck somewhere, what are the average turns for such conversation.
- how can you monitor medical hallucination which is common and severe in mental health medial advice recommendations
- can you use patient data (such as suicide patient medical records from MIMIC) to predict mental health using your language model

---

### Official Review · Reviewer_hfM9 · 2024-11-03

**Soundness:** 2
**Presentation:** 4
**Contribution:** 2
**Rating:** 3
**Confidence:** 4

**Summary:**

The paper introduces MentalArena, a self-play framework designed to train language models for diagnosing and treating mental health disorders. Due to privacy concerns and the scarcity of personalized mental health data, traditional methods struggle to build effective models in this field. MentalArena overcomes this challenge by allowing the language model to play both the roles of patient and therapist, thereby generating domain-specific personalized data. The framework consists of three main components: the Symptom Encoder, which simulates the cognitive and behavioral patterns of patients; the Symptom Decoder, which simulates interactions between patients and therapists by comparing diagnosed symptoms with encoded symptoms, thereby reducing intent bias; and the Model Optimizer, which collects diagnostic, treatment, and medication information from these interactions to fine-tune the model. The authors evaluated MentalArena on six benchmarks, showing that models fine-tuned using this framework perform significantly better in diagnosing mental health disorders compared to existing models, including GPT-4o.

**Strengths:**

The strength of MentalArena lies in its highly innovative self-play framework, which breaks away from the limitations of traditional prompt engineering commonly used to enhance language models for mental health applications. By enabling role-playing between patient and therapist, MentalArena dynamically generates data that simulates authentic interactions, allowing the model to learn and optimize autonomously through self-play—a promising exploration approach.

**Weaknesses:**

1. Lack of clear problem justification:

The authors claim to address the goal of "training language models for diagnosing mental health disorders," yet they evaluate the model using three general medical datasets (MedQA, MedMCQA, PubMedQA) that are unrelated to mental health. This approach fails to demonstrate the specificity of the model in diagnosing mental health disorders. After removing these general medical datasets, the model's performance on mental health-specific tasks is underwhelming, failing to show significant improvement in mental health tasks. This weakens the clarity and relevance of the paper's contribution to solving the stated problem.


| Model               | CAMS     | dreaddit | Irf      | AVG (Only mental health tasks)   |AVG (Including general medical datasets) |
|---------------------|----------|----------|----------|-------|----------------------------------------------|
| MentaLLaMa-13b      | 37.28    | 62.08    | 46.81    | 48.72 | 35.98 |
| Mental-LLM-alpaca   | 29.76    | 64.98    | 51.96    | 48.90 | 31.24 |
| Mental-LLM-t5       | 27.04    | 63.29    | 47.70    | 46.68 | 31.24 |
| GPT-4o              | 27.68    | 49.03    | 64.65    | 47.12 | 60.58 |
| GPT-4o+MedPrompt    | 31.52    | 53.27    | 64.65    | 49.81 | 64.22 |
| Base: GPT-3.5-turbo | 28.96    | 49.03    | 64.65    | 47.55 | 47.54 |
| +Chain-of-thought   | 29.92    | 49.03    | 64.65    | 47.87 | 48.87 |
| +MedPrompt          | 30.20    | 49.03    | 64.65    | 47.96 | 50.83 |
| +Ours               | 32.80    | 49.03    | 64.65    | 48.83 | 68.28 |
| Base: Llama-3-8b    | 25.12    | 58.45    | 45.76    | 43.78 | 54.75 |
| +Chain-of-thought   | 33.60    | 62.22    | 45.91    | 47.24 | 58.81 |
| +MedPrompt          | 35.08    | 61.59    | 48.05    | 48.24 | 60.17 |
| +Ours               | 29.60    | 65.46    | 52.25    | 49.77 | 61.39 |


2. Lack of domain expertise:

Although the authors claim that MentalArena is designed for the "diagnosis" of mental health disorders, the selected datasets do not fully support diagnostic tasks. Two of the so-called "diagnostic" datasets (such as Dreaddit and Irf) are primarily used for "assessment," meaning they measure symptoms and cognitive states rather than providing definitive diagnoses. Furthermore, the mental health datasets used are limited to depression, neglecting other important mental health disorders, such as anxiety disorders, bipolar disorder, and schizophrenia. The lack of evaluation on these key disorders limits the model's breadth and efficacy, making it inadequate for comprehensive validation in mental health diagnosis.

Overall, the paper overstates its contributions by positioning MentalArena as a diagnostic tool for mental health, despite limited evidence of effectiveness on mental health-specific tasks and a lack of comprehensive validation across diverse mental health disorders.

**Questions:**

1. The paper mentions cost-effectiveness as a contribution. How did the authors determine that MentalArena is indeed cost-effective?

2. How was the quality of the generated treatment and medication data evaluated?

3. Since MentalArena aims to reduce intent bias in patient-therapist interactions, what metrics or qualitative analyses were employed to confirm that intent bias was actually reduced?

4. How does the model handle ethical considerations, such as potential misdiagnoses or inappropriate treatment suggestions, especially since it is aimed at the sensitive domain of mental health?

5. On the dreaddit and Irf datasets, different methods (such as the base GPT-3.5-turbo, Chain-of-thought prompting, MedPrompt, and the Ours) show identical scores. What could be the reason behind this phenomenon?

6. Could additional mental health datasets (more disorders) from Table 1 [1] be tested? Also, since MedQA includes a Psychiatry subset, could results be tested on this as well?

[1] A Comprehensive Evaluation of Large Language Models on Mental Illnesses https://arxiv.org/pdf/2409.15687

---

### Note · Authors · 2024-11-19

I have read and agree with the venue's withdrawal policy on behalf of myself and my co-authors.